# Evaluation of Electrical Impedance Spectra by Long Short-Term Memory to Estimate Nitrate Concentrations in Soil

**DOI:** 10.3390/s23042172

**Published:** 2023-02-15

**Authors:** Xiaohu Ma, Luca Bifano, Gerhard Fischerauer

**Affiliations:** Chair of Measurement and Control Systems, Faculty of Engineering Science, University of Bayreuth, 95440 Bayreuth, Germany

**Keywords:** nitrate concentration, electrical impedance spectroscopy, EIS, recurrent artificial neural network, RNN, long short-term memory, LSTM, random forest

## Abstract

Monitoring the nitrate concentration in soil is crucial to guide the use of nitrate-based fertilizers. This study presents the characteristics of an impedance sensor used to estimate the nitrate concentration in soil based on the sensitivity of the soil dielectric constant to ion conductivity and on electrical double layer effects at electrodes. The impedance of synthetic sandy soil samples with nitrate nitrogen concentrations ranging from 0 to 15 mg/L was measured at frequencies between 20 Hz and 5 kHz and noticeable conductance and susceptance effects were observed. Long short-term memory (LSTM), a variant of recurrent artificial neural networks (RNN), was investigated with respect to its suitability to extract nitrate concentrations from the measured impedance spectra and additional physical properties of the soils, such as mass density and water content. Both random forest and LSTM were tested as feature selection methods. Then, numerous LSTMs were trained to estimate the nitrate concentrations in the soils. To increase estimation accuracy, hyperparameters were optimized with Bayesian optimization. The resulting optimal regression model showed coefficients of determination between true and predicted nitrate concentrations as high as 0.95. Thus, it could be demonstrated that the system has the potential to monitor nitrate concentrations in soils in real time and in situ.

## 1. Introduction

Nitrate nitrogen (nitrate-N) is one of the most important nutrients for securing the yield and quality of harvested products. However, excessive content of nitrate in soil leads to environmental problems. In Germany, nitrate is the main source of pollution for the groundwater. Nationwide more than a quarter of all near-surface groundwater fails to meet the nitrate target (50 mg/L) of the European Water Framework Directive [1]. Currently, nitrate concentrations in soils are measured by laboratory methods, such as flow injection analysis [2] and spectrophotometry [3]. Such methods require expensive equipment and a time-consuming collection and preparation of samples. This is why the methods are not used for continuous measurements, but only for sampling at larger intervals. To date, a reliable low-cost and in situ nitrate monitoring system is still not available and much needed [4,5].

A method that could eventually lead to such a monitoring system is electrical impedance spectroscopy (EIS). EIS is a powerful measurement technique used to measure the complex impedance of materials and systems over a range of frequencies. A small A-C signal stimulus is applied to a conductive electrode, thus obtaining a characteristic response that can be observed on the system surface. It has a high potential for fast, inexpensive, on-site measurements and is therefore used in many fields, such as material characterization [6], bioimpedance [7], battery diagnosis for state of health [8] and state of charge [9]. It is also very common for quantitative measurement of the ion concentrations in electrochemical sensors [10,11]. Its potential applicability to the soil monitoring problem is justified by the fact that soil is a three-phase system, composed of the solid, liquid, and gaseous phases. This results in a dielectric constant, or relative permittivity, of between 2 and 14, depending on the soil. The liquid phase is an ion solution containing nutrients, such as nitrate, necessary for soil fertility and has a high static dielectric constant, which is strongly influenced by the ion concentration—the higher the ion concentration, the smaller the dielectric constant [12,13]. Water is a polar liquid with high dielectric relaxation and therefore has a high static dielectric constant of 81 at room temperature. With the increase in ion concentration, the ions orient the water molecules around them, thereby reducing the dielectric constant to values as low as 40 by a local high-field effect [13].

In an electric A–C field, energy is absorbed and stored in soil solution mixtures. The effect of energy storage is connected with the real part $\varepsilon^{\prime }(\omega )$ of the permittivity, while the energy absorption is connected with the imaginary part $\varepsilon^{\prime \prime }(\omega )$ of the permittivity. Here, ω denotes the angular frequency.

Figure 1 shows a schematic representation of dielectric losses in soil as a function of frequency [14]. Various effects contribute to the complex dielectric constant. It is obvious that the ionic conductivity and the double-layer effect play major roles in the imaginary part of the soil permittivity at lower frequencies. Both effects are strongly correlated to the ion concentration [15]. At the same time the double-layer effect leads to a frequency-dependence, which also influences the real part of the permittivity [15].

When there is more than one ion species present in the soil, they all contribute to the impedance spectrum. The mixture problem is always a challenge in chemical sensing [16]. Here we do not deal with it, but assume that only nitrate ions, if any, are present in the soil. In the long run, one will want to remove this simplification, but in a first step it may be justified by the fact that by far the biggest soil problem is precisely nitrate ions and not any other ions [17].

With this assumption, in principle, these relationships in Figure 1 allow one to estimate nitrate concentrations in soil indirectly from measured impedance spectra of soil samples. In practice, however, there is no one-to-one mapping between given physical causes and single features of an impedance spectrum. There exists a vast literature on physics-based models of impedance spectra, mostly relying on equivalent circuits [18,19,20]. The spectrum of a distributed system (with its uncountably infinite number of degrees of freedom) can only be approximated by a lumped-element circuit (with its finite number of degrees of freedom). The approximation may be made arbitrarily good by increasing the number of circuit elements. Alternatively, one can use elements with frequency-varying characteristics, such as Warburg impedances [21] or constant-phase elements [22]. These approaches exacerbate the one-to-one mapping problem and usually do not allow for predicting the spectral response to physical changes in the system, such as geometry variations or analyte concentration changes. This remark applies even more to mere a-posteriori description methods, such as DRT [23,24] or to situations in which the entire spectrum cannot be measured for practical reasons (e.g., when measurement times are limited to perform a fast measurement such that small-frequency responses cannot be observed). However, even in the absence of a one-to-one relationship between a nitrate concentration and a single feature of a measured impedance spectrum, the desired information is somehow hidden in the spectrum and entangled with other pieces of information. EIS was used in [25] together with effective-media models to estimate nitrate concentrations in soils. The results demonstrate the potential of the approach. However, model-based methods rely on soil properties and ambient conditions, and the individual influences of water, nitrate, and soil composition still remain unclear. For this reason, the resulting error is less than satisfactory and one cannot state that the method has been fully established. In field environments, other than in laboratories, it is impossible to measure all relevant influence quantities or to isolate the measurement system from all the quantities. For reasons of effort, it is also impractical to describe all relevant effects. Such problems are best handled by machine learning approaches [26].

In this work, machine learning and neural networks are used to extract information from impedance spectra to explore how well they are able to untangle the information mess. This is consistent with the reasoning that we only want to gain information on the nitrate concentration in the soil, despite the many influence effects, but do not require physical cause-and-effect relationships (which, incidentally, are also not provided by the majority of parametrized models, such as equivalent circuits).

To be more concrete, we consider long short-term memory (LSTM), a variant of recurrent artificial neural networks (RNN) [27], to estimate nitrate concentrations in soils based on measured impedance data. LSTM RNNs are particularly useful to cope with long-term dependencies between different data points in time series (hence their name) [27,28]. They can detect patterns in a time series, such as a voice recording, because they not only consider the data values but also include the positions of the data points in the series in their prediction. There is no reason why the method should be restricted to time series and should not be applicable to arbitrary data sequences, such as impedance spectra. In fact, LSTM RNNs have been applied repeatedly to EIS data, e.g., to predict fuel-cell degradation [29] and estimate the state of healthy lithium-ion batteries [30]

Often, it is not clear a priori which measurement data features are best passed to a neural network to optimize its prediction accuracy. Before training the network (here: the LSTM), the most suitable feature subset must be identified by a feature selection algorithm. The resulting removal of redundant and irrelevant data facilitates the design and the training of the neural network without incurring much loss of information [31]. Redundant data may lead to overfitting and irrelevant data may cause long training times. In many studies [32,33,34] LSTM RNNs are combined with different feature selection algorithms. In [32], LSTM RNNs combined with random forest as feature selection methods (cf. below) are used to predict the battery degradation. In [33] an improved forward selection using light gradient boosting machines selected the optimal feature set and then LSTM RNNs served to predict photovoltaic power generation. In [34] the features are selected by Pearson’s correlation method with maximum relevancy and minimum redundancy as criteria and then the selected features are passed to LSTM RNNs to predict the short-term energy consumption of customers.

All these feature selection methods treat the series data first as discrete input data and then estimate the importance of each feature. The dependency between data points in the series is not considered. The selected features are delivered as sequence of data to LSTM RNNs in which the dependencies between data points play an important role [27]. Therefore, the feature selection methods mentioned above may not be optimal for LSTM RNNs. In this work, a new feature selection method, combining sequential forward selection and LSTM, is developed, which processes variable-length sequences of inputs to evaluate the relative importance of features in the measured data (electrical impedance spectra, soil properties, and ambient conditions). With selected features the hyperparameters of LSTM RNNs were tuned by Bayesian hyperparameter optimization and an optimized regression model was built to estimate the nitrate concentration in sandy soils.

To validate the performance of this new method and compare the evaluation result of nitrate concentration in synthetic soils with LSTM, feature selection and evaluation with random forest (RF) is applied for comparison, all in the context of processing and interpreting impedance spectra of soil samples.

RF is a known ensemble learning method that constructs a multitude of decision trees independently at training time. In particular, each RF decision tree is trained using a bootstrap set, which is randomly sampled from the input data with replacement (here the input data are impedance spectra and measured values of other quantities, such as soil density). In this way, about two thirds of the instances in the bootstrap set are unique and used to build a decision tree, and about one third are left out as out-of-bag (OOB) instances used to validate the decision tree [35]. The ensemble produces a forest of trees and outputs of all trees are aggregated to produce a final prediction. In classification problems, the final prediction is voted by the majority of trees. In regression, the output is the average of the individual tree predictions.

As all decision trees are built over a random extraction of the instances from the database and a random extraction of the variables, RF is also widely used to measure variable importance. A suitable metric for this is the mean decrease in impurity (MDI), also known as Gini importance, defined as the sum of all decreases in Gini impurity [36] due to a given variable when this variable is used to form a split in the RF. Another metric is the permutation importance which quantifies the variable importance based on the reduction in model accuracy when permuting the values of a variable included in a subset of variables that were split to grow the tree. Both metrics have been discussed and compared extensively [37,38,39]. The mechanism of MDI is biased, i.e., it tends to inflate the importance of continuous or correlated variables [37,39]. When the data set is too small, permutation importance is also unstable and may lead to a wrong ranking of variable importance [38]. Thus, the metric should be chosen carefully according to the measurement problem. It is not clear a priori which metric is most suitable for the selection of features from impedance spectra.

Another advantage of RF is the ability to measure variable association, which is an indication of correspondence between variables. Variable correspondence can increase the standard errors of a machine learning model [40]. It is an essential step to recognize the relationship between variables for establishing a robust and accurate machine learning model. RF is a tree-based algorithm and is able to use surrogate splits to measure the association between variables. Such a surrogate split is an alternative to the optimal split at a given node in a decision tree. It uses a similar or correlated variable and split criterion to replace the optimal split variable and split criterion [41].

To the best of the authors’ knowledge, this work enhances the state of the art in the following respects:The method of measuring nitrate concentration in soil based on the combination of EIS with LSTM networks is characterized for the first time.The advantage of a new feature selection method over conventional methods, especially random forest, is demonstrated for the first time, the advantage being a higher coefficient of determination between the actual nitrate concentration and the concentration predicted by the LSTM network fed with the selected features.The results presented improve the quantitative understanding of the potential of the described approach in the context of the low-cost, robust, fast, in situ, and soil property-independent measurement of nitrate concentrations in soils.

## 2. Materials and Methods

As material under test (MUT), dry quartz sand from Karl Inzelsberger Sand and Kaolinwerk in Creußen, Germany, prepared with six different nitrate-N concentrations, viz., 0, 2, 4, 6, 8, and 10 mg/kg, was used as synthetic soil. These nitrate-N concentrations are below the recommended levels of 20 to 40 mg/kg in soils depending on crop [42]. A total of 500 g of pure quartz sand was enriched in a beaker with sodium nitrate solutions prepared according to international standards [2]. After mixing and drying, the sand–nitrate mixture was examined at different water contents (the deionized water concentrations were between 0 and 150 g/kg). The water content here is the ratio of the mass of water and mass of dry mixture. Dry mixture mass is expressed by the mass of mixture dried (up to the constant weight) at 105 °C. In agriculture, the mass water content is not used frequently but a necessary step for the measurement of volumetric water content with gravimetric methods, as will be seen later. In the field, the volumetric water and nitrate-N contents offer more information about the ratio of pores filled with water and make it easier to calculate the quantity of water and nitrate in an area. Thus, the above-mentioned water and nitrate-N mass contents were converted to volumetric contents by:(1)$$\theta_{W}=\frac{m_{W}}{V}=\frac{m_{S}\cdot \omega_{W}}{V}=\rho_{S}\cdot \omega_{W}$$
(2)$$\theta_{N}=\frac{m_{N}}{V}=\frac{m_{S}\cdot \omega_{N}}{V}=\rho_{S}\cdot \omega_{N}$$
where $\theta_{W}$ and $\theta_{N}$, respectively, are the volumetric water and nitrate-N contents, $\omega_{W}$ and $\omega_{N}$ are the corresponding mass contents, $m_{W}$, $m_{N}$ and $m_{S}$ are the masses of water, nitrate-N, and sand in the measurement volume *V*, and $\rho_{S}$ is the density of dry sand in the measuring cell. Obviously, even when the mass content remains constant, the vol–metric content can vary.

To investigate various synthetic soil mixtures, the measuring cell shown in Figure 2 was used. As polytetrafluoroethylene (PTFE) is a high-temperature-resistant non-wetting material and stainless steel is stable to nitrate solutions, the measuring cell consists of two opposing square plates made of PTFE with a side length of 15 cm and two circular stainless-steel electrodes with a diameter of 13 cm centered in them, forming a plate capacitor with an electrode spacing of 1 cm. Both plates are removable so they can be easily cleaned after each measurement of the previous sample. The MUTs were filled into the space bit by bit between the electrodes until the material was flush with the top edge of the measuring cell. Care was taken to keep the bulk density as planned.

The impedance $\underline{Z}=\left|\underline{Z}\right|e^{j\varphi }=R+jX$ was measured over a frequency range from 20 Hz to 5 kHz using an Agilent E4980A impedance meter (98 measurement points per frequency sweep). For electrochemical systems, lower frequencies (as low as 1 mHz) would provide more information but would also take more time for the measurement. Therefore, as we are interested in fast measurements over larger areas (agricultural fields), we restricted the frequency to values above 20 Hz. At each frequency, both the apparent impedance |*Z*| and the impedance phase φ were recorded and then evaluated in Matlab R2021a. Each MUT was repeatedly measured 20 times.

Figure 3a shows the measured frequency-dependent conductance G=Re{Y_}=Re{1/Z_} as a function of nitrate-N and water concentrations. *G* increases with increasing nitrate-N and water contents, the nitrate-N sensitivity being significantly greater than the moisture sensitivity. This demonstrates the potential for monitoring the nitrate-N concentration in soil with water as influence quantity, but also emphasizes the need for prior calibration.

Likewise, Figure 3b shows the measured frequency-dependent susceptance B=Im{Y_}=Im{1/Z_} as a function of nitrate-N and water concentrations. In the low-frequency region from 20 Hz to 1 kHz, *B* increases with increasing nitrate-N and water contents. This can be explained by the electrical double-layer effect, which is strongly related to the ion concentration [15]. It may provide a possibility to monitor the nitrate-N concentration at low frequencies.

The data in Figure 3 suggest that the designed impedance sensor can measure the nitrate-N concentration, possibly with prior calibration and possibly in the field (rather than the laboratory). As calibrations to take into account specific soil types and water content are likely to require too much time and effort, we looked for ways to remove the need for calibration. The reasoning is about the following: impedance spectra contain an abundance of signal features. This richness of information should allow one to estimate both analyte concentrations and influence quantity values (soil density, water content, etc.). The estimation must be automated, which calls for a machine learning algorithm.

Before the analysis using machine learning, it is an important step to prepare data, which involves reformatting data, making corrections to data and enriching data. The raw data sampled from our experiments consist of measured impedances $\underline{Z}(f)$ (2520 spectra from 126 synthetic soils compositions, each spectrum comprising 98 discrete frequencies) and physical properties of soils for each sand composition (density $\rho_{b}^{}$, volumetric water content $\theta_{w}^{}$). The measured impedance was transformed into conductance *G* and susceptance *B* to provide a better view of the electrochemical changes in the measuring cell. As shown in Figure 3, *G* and *B* are functions of frequency and the derivatives of them may contain useful information. Therefore discrete approximations of dB/df, dG/df, $d^{2}B/df^{2}$, and $d^{2}G/df^{2}$ as additional features are added to test whether they can facilitate the machine learning. As LSTM RNN process sequences of data, the features passed to it were *G* and *B* (vectors of length 98), $\rho_{b}\cdot a$ and $\theta_{w}\cdot a$ (where $a={\left(1,1,\ldots ,1\right)}^{T}$ is a size-98 column vector of 1s), and the derivatives just discussed (also padded to a length of 98). The padding of the scalars or shorter vectors to the length of the impedance spectrum (98) does not add information but is needed to pass data to all neurons of the RNN input layer. The features passed to RF were frequency-discrete. Before passing the features, they were centered and scaled to have zero mean and unity standard deviation.

## 3. Results

### 3.1. Feature Selection

In the following, feature selection by RF and by a newly developed method based on LSTM is evaluated with respect to the requirements for nitrate concentration measurement.

#### 3.1.1. Feature Selection by Random Forest

RF provides estimates both of the association between various features and of the feature importance. The association and importance of the eight features available in this work are respectively presented in Figure 4a,b. The importance was calculated using permutation importance as metric.

Figure 4a confirms that the association between distinct features is small, which indicates that the features are independent. The bar graph in Figure 4b reveals that the conductance *G* is the most important feature, followed by the volumetric water content $\theta_{w}^{}$ and the soil density $\rho_{b}^{}$. Other features have but a negligible effect on the nitrate concentration estimation. Similar results have been obtained with the sequential feature selection method combined with a support vector machine.

#### 3.1.2. Feature Selection by Sequential Forward Selection (SFS) Based on LSTM

Since feature selection with RF does not consider the dependencies between different frequency points, it may not provide the optimal result when training LSTM RNN. We therefore developed a new feature selection method combining SFS with LSTM to select proper features of impedance spectra and their derivatives.

SFS selects features sequentially based on a criterion to be minimized over all feasible feature subsets. In our case, the validation root mean square error (RMSE) of the LSTM network output was used as criterion. In SFS the features are sequentially added to an empty candidate set until the addition of further features no longer improves (i.e., decreases) the RMSE by more than a preselected minimum value, set to 0.1 mg/L in the given context.

The difficulty of using the RMSE of the LSTM output as a criterion for SFS is that the optimum hyperparameters of the LSTM network vary with the feature subsets. To solve this problem, Bayesian hyperparameter optimization was used to find out the optimum hyperparameters for the LSTM RNNs of all feature subsets. Bayesian optimization calculates the optimum hyperparameters in a specified range of values and uses a metric (in our case, the RMSE of the LSTM RNNs) to optimize. In this way, not only the optimum hyperparameters are determined, but also the RMSE for the SFS process. For every subset, an LSTM RNN with optimized hyperparameters and minimized RMSE was trained. The one feature that decreases the RMSE most at each step is retained in the further process.

The main structure of an LSTM RNN is shown in Figure 5. A dropout layer after each LSTM layer is used to avoid overfitting. The dense layers are to reduce the dimension of the output from previous layers to provide a smooth transition from the hidden layers to the output layer. The ReLulayer just before the output ensures a non-negative output. 

The hyperparameters listed in Table 1 were optimized inside predefined intervals. The feature selection results with SFS based on LSTM, the optimal hyperparameters for each step are presented in Table 2. The whole feature selection time for step 1 through step 4, shown in the table, took 32.8 h on a workstation with a 3.7 GHz quad core and a memory of 64 GB. In the first step, the susceptance *B* decreases the RMSE most and thus is the most important feature for a successful LSTM estimator. Adding the volumetric water content $\theta_{w}$ decreases the RMSE more than the remaining features. Similarly, the third and fourth most important features are the conductance *G* and the soil density $\rho_{b}$. Adding other features had no appreciable effect on the target criterion. In particular, the first and second derivatives of *G* and *B* are not needed to estimate nitrate concentrations in soils.

#### 3.1.3. Comparison

One notices that the feature selection results from RF on the one hand and SFS based on LSTM on the other hand are not compatible. The most important feature according to the latter, the susceptance *B*, had little or no effect according to the former. Conversely, the most important feature by RF, the conductance *G* (cf. Figure 4), is only ranked third by SFS. Both selection methods assessed the volumetric water content as crucial and soil density as less important. The reason for the differences in feature ranking and the consequences for the nitrate concentration measurement accuracy are discussed below.

### 3.2. Experimental Results for Nitrate Concentration Measurement

In order to quantify the influence of the features on the estimation accuracy, the feature selection results were validated by the regression models from their own algorithms. The regression models built from RF take an input vector containing the measured quantities at one specific frequency as well as the numerical value of said frequency. When building regression models from RF and LSTM RNNs, our impedance measurements provided 2520 spectra. They were split into training and validation data in the ratio of 80:20. After the training, the models were tested using 360 spectra unused up to this point. Both algorithms took less than 10 min to train regression models and the inference times for both were around one second. The quality of the prediction results was judged by the coefficient of determination between true and estimated nitrate-N concentration, defined as:(3)$$R^{2}=1-\frac{\sum_{i}{\left(Y_{test-i}-Y_{pre}{}_{-i}\right)}^{2}}{\sum_{i}{\left(Y_{test-}{}_{i}-\bar{Y_{test}}\right)}^{2}}$$ Here, i is the case index, $Y_{test-i}$ and $Y_{pre}{}_{-i}$ are the actual and predicted nitrate-N concentrations of the test data, respectively. $\bar{Y_{test}}$ is the mean of the actual concentration.

Figure 6 and Figure 7 illustrate the quality of the nitrate-N concentration estimation by the various neural networks with different feature subsets. Using but one feature (the RF-favored conductance spectrum *G* or the SFS-favored susceptance spectrum *B*, respectively), one obtains results between rather unconvincing, Figure 6a, and quite good, Figure 7a. Adding other features step by step, as suggested by Figure 4b and Table 2, respectively, improved the estimation accuracy monotonically. In all cases, the LSTM estimator based on SFS performed better than the RF estimator, although the differences tend to become smaller, the more features are passed to the estimators. The optimum feature combination turned out to be *B*, $\theta_{w}$, and *G* (in other words: the full complex impedance spectrum *Z* enhanced by the additional information $\theta_{w}$). This feature set led to an excellent measurement result with $R^{2}=0.95$, Figure 7c, which could not be improved further by adding more features, such as soil density $\rho_{b}$, Figure 7d. The additional features might prove helpful for the generalization of this approach to other soils, which was not part of this work.

It is obvious that the RF-selected features are not the optimal choice for the LSTM RNN. In fact, a regression LSTM RNN trained with the top three RF-favored features (*G*, $\theta_{w}$, and $\rho_{b}$) resulted in a coefficient of dermination of only $R^{2}=0.82$ instead of $R^{2}=0.95$. Both feature selection methods work well with regression models based on their own algorithms, but cannot be used interchangeably. This implies that a feature selection method based on point data, such as RF, may not be optimal for regression models based on vector-valued input data with dependencies between the vector elements (time or frequency series). With respect to our impedances, such a behavior is already noticeable in Figure 3. There is an unambiguous relationship between the measured conductance *G* and the nitrate concentration at all frequencies, Figure 3a. In contrast, the susceptance *B* depends less and less on nitrate concentration with increasing frequency (and in practice, measurement errors would prevent one from determining nitrate concentration from high-frequency susceptance). However, the LSTM method takes the whole frequency spectrum as input, and as *B*(*f*) depends more strongly on *f* than *G*(*f*) and thus contains more information as sequence input, LSTM can estimate the nitrate concentration better from *B* than from *G* and better than the RF-based regression model.

### 3.3. Literature Comparison

As discussed in the introduction, the result in [25] using EIS with effective-media models is less satisfying because of the dependency on soil conditions. In addition, the measuring range of nitrate-N is from 14 mg/L to 70 mg/L which can hardly meet the field requirements. Another attempt was made to monitor the nitrate concentration in soils in situ without pretreatment using spectrophotometry in [45]. This system is strongly dependent on the water content in the soil and is not able to measure the nitrate over large area. Moreover, the measuring range of nitrate-N used in [45] is from 0 to 500 ppm, which is far beyond the field requirements. Through comparison, our work shows the best possibilities to overcome these problems. We hope to develop a system based on it that can be mounted on a tractor and allow measurements across large agriculture areas.

## 4. Conclusions

Based on synthetic soil samples, we have demonstrated that machine learning methods can extract nitrate-N concentrations in soil from measured impedance spectra enhanced by selected additional physical variables with errors that are compatible with field requirements, and the demonstrated measurement accuracy was far below the recommended nitrate-N concentration in soils.

It was also shown that proper feature selection yields objective figures of merit for the impact of different features. Based on this, one can design RNNs that estimate the nitrate concentration in soils robustly and accurately against the influence of water content and soil density. Feature selection by SFS based on LSTM improved the nitrate concentration estimation accuracy of an LSTM RNN to $R^{2}=0.95$, which is better than the values obtained by more common feature selection. This result was obtained when the complete complex impedance spectrum and the water content of the soil were passed as input data to the neural network. Adding the soil density as additional input data did not improve the prediction quality any further. Therefore, in practice, soil density is not expected to influence field measurements significantly.

Of course, in the field one will be faced with many types of soil (not just sandy as in our experiments) and with additional influence quantities. Our further goal is to conduct more experiments to expand the measuring range and include more soil types, such as potting soils, brown soils, and clay mixtures. In this way, we hope to implement a method based on machine learning that is capable of measuring nitrate-N concentrations virtually independently of soil properties and types. True field measurements cannot rely on bulky and expensive laboratory equipment but will require application-specific printed-circuit-board impedance meters.

## Figures and Tables

**Figure 1 sensors-23-02172-f001:**
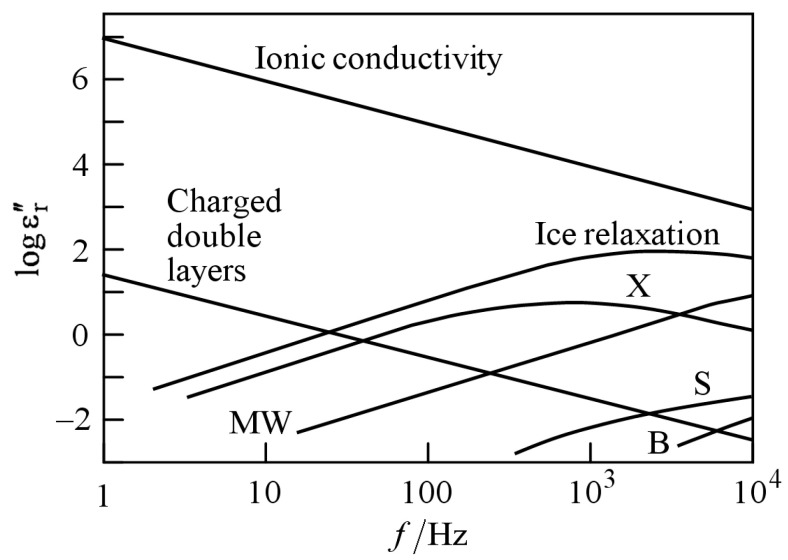
Contributions to the imaginary part $\varepsilon_{r}^{\prime \prime }=\varepsilon^{\prime \prime }/\varepsilon_{0}$ of the relative permittivity of soil (after [14]). B: bound water relaxation; MW: Maxwell–Wagner effect; S: surface conductivity; X: crystal water relaxation.

**Figure 2 sensors-23-02172-f002:**
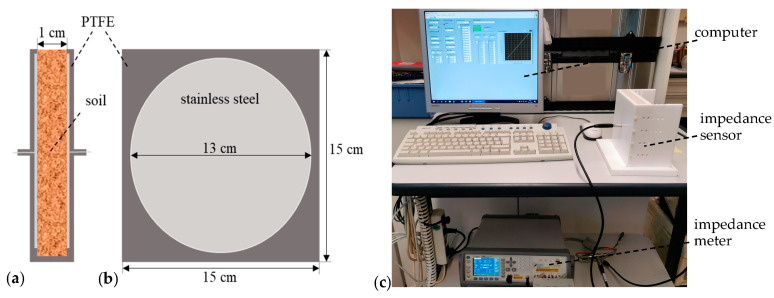
Parallel-plate impedance sensor. (**a**) Cross section. (**b**) View of the right plate from the left. (**c**) Photograph of the setup with impedance meter and computer.

**Figure 3 sensors-23-02172-f003:**
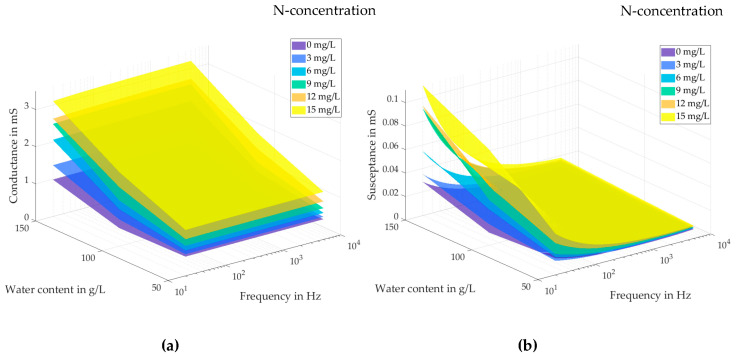
Measured test cell conductance (**a**) and susceptance (**b**) as a function of the nitrate-N concentration (in mg/L) and of the water content (in g/L) in synthetic soil (quartz sand), and of frequency, reprinted from Ref. [43].

**Figure 4 sensors-23-02172-f004:**
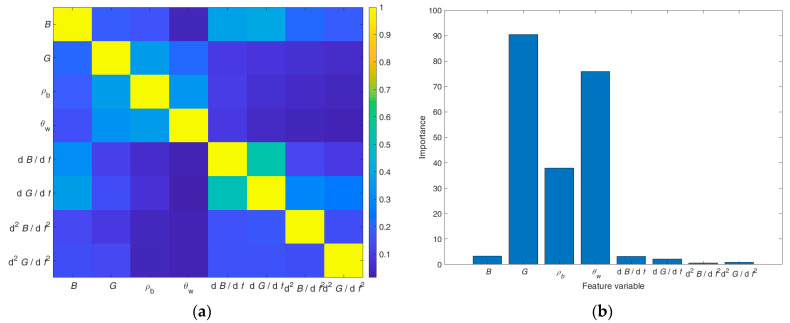
Feature association estimation (**a**) and importance estimation (**b**) using RF, reprinted from Ref. [44].

**Figure 5 sensors-23-02172-f005:**
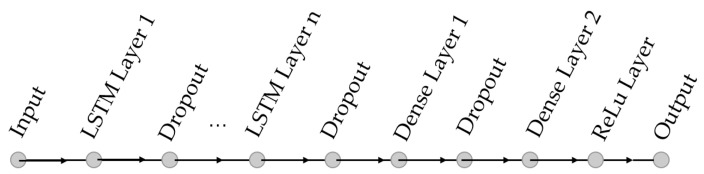
LSTM network structure.

**Figure 6 sensors-23-02172-f006:**
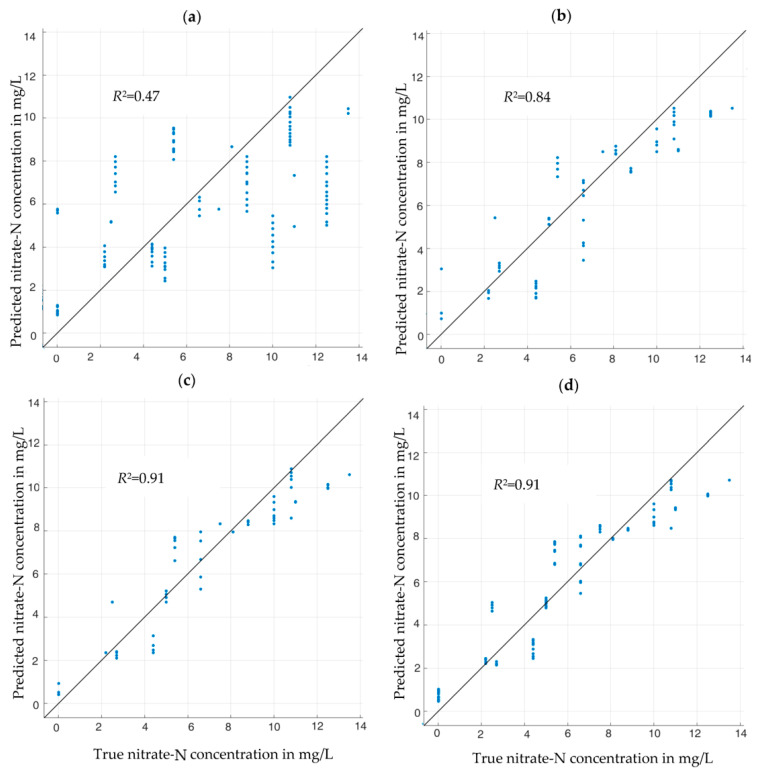
Nitrate-N concentrations predicted by RF-built regression models from measured impedance spectra of synthetic soil samples plus selected other physical variables. (**a**) Results based on the conductance spectrum *G* as feature. (**b**) Results with features *G* and $\theta_{w}$. (**c**) Results with features *G*, $\theta_{w}$, and $\rho_{b}$. (**d**) Results with features *G*, $\theta_{w}$, $\rho_{b}$, and *B*. Blue dots: estimation results for each of the 360 validation data sets. Straight lines: goal (predicted value = actual value). The *R*^2^ values shown in each chart are the coefficients of determination.

**Figure 7 sensors-23-02172-f007:**
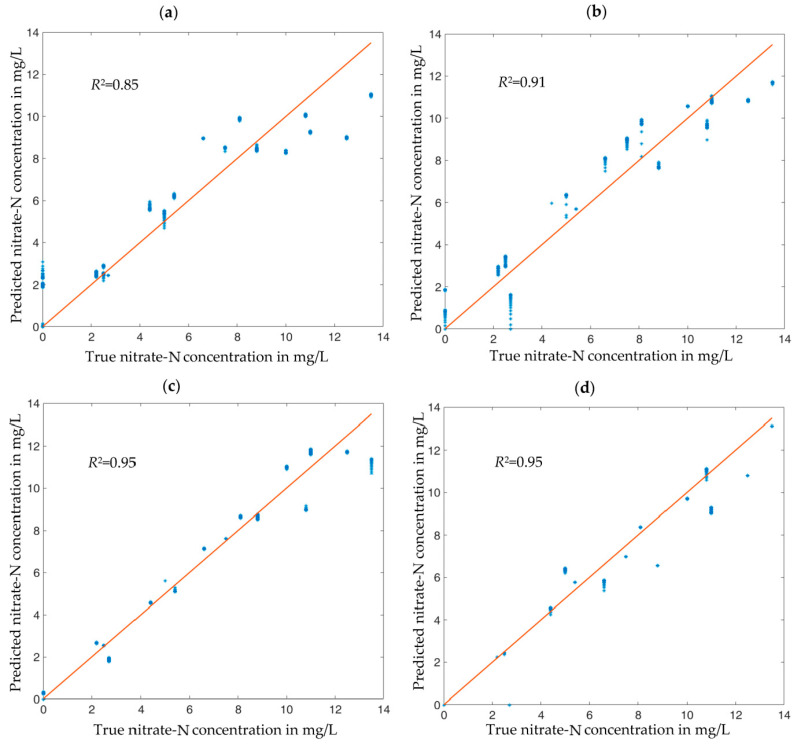
Nitrate-N concentrations predicted by LSTM-built regression models from measured impedance spectra of synthetic soil samples plus selected other physical variables. (**a**) Results based on the susceptance spectrum *B* as feature. (**b**) Results with features *B* and $\theta_{w}$. (**c**) Results with features *B*, $\theta_{w}$, and $\rho_{b}$. (**d**) Results with features *B*, $\theta_{w}$, $\rho_{b}$, and *G*. Blue dots: estimation results for each of the 360 validation data sets. Straight lines: goal (predicted value = actual value). The *R*^2^ values shown in each chart are the coefficients of determination between true and estimated concentrations.

**Table 1 sensors-23-02172-t001:** Hyperparameters to be optimized using Bayesian optimization with predefined parameter ranges.

No. of Hyperparameter	Description	Range	Type
1	Number of LSTM layers, *N*	1…3	Integer
2	Numbers of neurons in each hidden layer	(1…100) × *N*	Integer × *N*
3	miniBatchSize ^1^	2…20	Integer
4	initialLearnRate ^1^	10^−3^…10^−1^	Real

^1^ Nomenclature of MATLAB R2021a Deep Learning Toolbox.

**Table 2 sensors-23-02172-t002:** Results of feature selection by SFS with the validation RMSE of the LSTM output as criterion to be minimized.

Step	Features	Validation RMSE, in mg/L	Optimum Value of Hyperparameter No. (cf. Table 1)			
			1	2	3	4
1	*B*	2.1920	2	(25,75)	18	0.005
2	*B* and $\theta_{w}$	1.2364	2	(60,12)	16	0.007
3	*B*, $\theta_{w}$, and *G*	0.6873	2	(89,99)	12	0.001
4	*B*, $\theta_{w}$, *G*, and $\rho_{b}$	0.1431	2	(77,45)	16	0.001

## Data Availability

The data presented in this study are openly available at https://doi.org/10.15495/do_ubt_2067, accessed on 5 February 2023, provided by the authors.

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
