# Peer review of "Evaluation of Electrical Impedance Spectra by Long Short-Term Memory to Estimate Nitrate Concentrations in Soil"

_sensors, 2023, doi:10.3390/s23042172_

Round 1

Reviewer 1 Report

the manuscript is reviewed. the comments are 

related work and introduction section can be merged. the present indroduction section is very long. it should be concise. 

what is the source of quartz sand ?

Did the author check the chemical composition of the soil after preparing the mixture ?

author should expand PTFE in line220.

authors are citing other reference for figure 3. Is it produced by author ? I feel the manuscript is original article. 

conclusion section is the summary of the outcome. author should not include reference in conclusion. 

Author Response

Thanks for your kindly review and the positive response. Below are the solutions based on your comments and suggestions:

  1. Introduction and related work are merged together. Because of this action, the green texts in the new manuscript (please see the attachment) were same from the old manuscript but relocated.
  2. Source of quartz sand is addressed in line 312-313 in the new manuscript.
  3. No, we didn’t check the chemical composition of the soil after preparing the mixture. But the nitrate concentrations of the mixtures were also measured with Flow Injection Analysis method. We didn’t mention it in our script because the results cannot be compared directly. The reason may be that, in our measurement cell 350-gram sand can be filled. The Flow Injection Analysis takes only around 30-gram sand as sample. In other word: FIA measures locally, EIS measures integrally.
  4. PTFE was expanded in line 335.
  5. Yes, the figure has been produced by us, but has already been published in reference [43] as indicated in the figure caption.
  6. The reference in the conclusion was removed to line 314-315 in the new manuscript.

Reviewer 2 Report

The article is very interesting. But what is the practical aspect of this research. How will this affect field research?

Were the nitrate concentration results compared with the spectrophotometric method of nitrate determination used?

Author Response

Thanks for your kindly review and the positive response.

  1. The practical aspect of this research is addressed in line 558-559(please see the attachemnt).
  2. No, we didn’t compare our results with the spectrophotometric method. But we compared the nitrate concentration results with the Flow Injection Analysis method. We didn’t mention it in our script because the results cannot be compared directly. The reason may be that, in our measurement cell 350-gram sand can be filled. The Flow Injection Analysis takes only around 30-gram sand as sample. In other word: FIA measures locally, EIS measures integrally.

Reviewer 3 Report

The paper proposed machine learning methods to predict nitrate-N concentrations in soil. The experiment on real data shows the effectiveness of the two proposed regression models and the feature selection module. Overall, the paper is well-organized and easy to follow. However, the paper has several points that need attention as follows:

1. Novelty may be limited. Both random forest and LSTM, as popular machine learning methods, have been commonly used by other researchers in both feature selection and prediction. It would be better if authors can point out the uniqueness of the work more explicitly.

2. Are the selected features data-specific? Since the features are selected based on the current training data, will they be generalized well to different soil types? It will be interesting to discuss the data/material-dependency of the feature selection.

3. In terms of the performance evaluation, it would be better if the authors can make a comparative study with the start-of-art works.  

4. For the system evaluation, authors may need to provide information on the training cost (training time, computation cost, etc.) and the inference time.

Author Response

Thanks for your kindly review and the positive response.

  1. A bit more explanation of the uniqueness is addressed in line 159-164.
  2. Yes, the selected features are data-specific. We added a bit explanation in line 509-511. We also mentioned in line 579-580 that we are now actually investigating our measurement method in different soils, more precisely in potting soils, brown soils, and clay mixtures. The data/material-dependency will be discussed in future work.
  3. The comparison with state-of-art works is addressed in a new section 3.3 in line 548-558.
  4. We added text with details on the time required for feature selection in line 460-462, training and inference times in line 490-492.

Round 2

Reviewer 1 Report

the author addressed all the comments.

Reviewer 3 Report

The authors have addressed all my previous concerns in the revised manuscript.